# A 0.6 *V_IN_* 100 mV Dropout Capacitor-Less LDO with 220 nA *I_Q_* for Energy Harvesting System

**DOI:** 10.3390/mi14050998

**Published:** 2023-05-03

**Authors:** Yuting Zhang, Qianhui Ge, Yanhan Zeng

**Affiliations:** 1School of Electronics and Communication Engineering, Guangzhou University, Guangzhou 510000, China; iyuting_2002@163.com (Y.Z.); 2112130007@e.gzhu.edu.cn (Q.G.); 2Key Lab of Si-Based Information Materials & Devices and Integrated Circuits Design, Guangzhou University, Guangzhou 510000, China

**Keywords:** low-dropout regulator, ultra-low-input voltage, high power efficiency, bulk modulation, energy harvesting

## Abstract

A fully integrated and high-efficiency low-dropout regulator (LDO) with 100 mV dropout voltage and nA-level quiescent current for energy harvesting has been proposed and simulated in the 180 nm CMOS process in this paper. A bulk modulation without an extra amplifier is proposed, which decreases the threshold voltage, lowering the dropout voltage and supply voltage to 100 mV and 0.6 V, respectively. To ensure stability and realize low current consumption, adaptive power transistors are proposed to enable system tropology to alter between 2-stage and 3-stage. In addition, an adaptive bias with bounds is utilized in an attempt to improve the transient response. Simulation results demonstrate that the quiescent current is as low as 220 nA and the current efficiency reaches 99.958% in the full load condition, load regulation is 0.0059 mV/mA, line regulation is 0.4879 mV/V, and the optimal PSR is −51 dB.

## 1. Introduction

Energy harvesting (EH) technology converts weak energy harvested from the environment into electricity [1,2,3]. EH has been used in many fields, such as the Internet of Things (IoT), wearable devices, and wireless sensors [4,5,6]. The output voltage collected from energy harvesting is unstable. Therefore, the low-dropout regulators (LDOs) are used to provide a stable power supply for the loads, as they have the advantages of uncomplexity, a low cost, and being noise-immune.

A typical EH system is depicted in Figure 1. Considering the weakness of the energy source and low collection efficiency, energy collection suffers from a low supply voltage and high power consumption [7,8,9]. Consequently, new challenges and requirements are set forth for the supply voltage and power conversion efficiency of LDO.

For the purpose of improving the integration and reducing the cost, [10,11,12,13] proposed the output-capacitor-less (OCL) LDOs consuming μA-level static currents. However, low-power equipment should consume as little electricity as possible during standby and operation. To further reduce the current to nA levels, [14] adopts an adaptive bias solution based on the super-source follower (SSF) structure. Nevertheless, the minimum supply voltage is as high as 1.8 V, and struggles to meet the EH requirement. In addition, the large dropout voltage leads to large power loss.

Another approach to improving efficiency is to reduce the dropout voltage [15]. A dropout voltage can be defined as the product of the valid channel resistance of the component and the load current in non-regulatory conditions [16]. In [17], a large and wide ratio power transistor is proposed to reduce the equivalent resistance, but this requires a large area. One method to reduce the threshold voltage is the floating meter [18,19], and the other method is body bias [20,21,22]. Recent research has shown that the threshold voltage can be reduced using current-driven bulks [23]. Since $V_{B}$ is less than $V_{S}$ in the PMOS power transistor, the source-end bipolar crystal pipe can be turned on to reduce the threshold voltage. This extra adaptively biased current-driven loop generates a lot of static current, which cannot meet the low $I_{Q}$ requirements.

This paper proposes a 100 mV-dropout and high-efficiency OCL-LDO with low supply voltage, which uses bulk modulation and adaptive bias technology with adaptive power transistors (APT). The bulk modulation without using the auxiliary amplifier has significantly reduced the dropout voltage to 100 mV. The adaptive bias can quickly detect the transient jumps to shorten the recovery times and reduce the quiescent current. Through the use of APT, the system is able to switch between 2-stage and 3-stage amplifiers under different load conditions, ensuring its stability in the full load range. Further, APT reduces power consumption by reducing unnecessary current consumption. On the one hand, ultra-low $V_{IN}$ and $V_{OUT}$ are suitable for weak EH systems, since they require a low input voltage and output voltage. On the other hand, by reducing the dropout voltage, high efficiency can be achieved with low power consumption.

In the rest of this paper, the operating principle of the proposed LDO is discussed in Section 2. Section 3 and Section 4 describe the circuit implementation and stability analysis, respectively. The simulation results, along with comparisons with the state-of-the-art designs, are reported in Section 5. Finally, conclusions are drawn in Section 6.

## 2. Operation Principle

### 2.1. Bulk Modulation without Amplifier

The amount of energy collected by the EH system is extremely small. DC-DC Boost converter only produces a few hundred mV of output voltage, requiring a low $V_{IN}$ in the LDO design. Generally, the $V_{IN}$ can be expressed as:(1)$$\begin{array}{c} V_{IN}=V_{G}+\left|V_{GSP}\right|\geq V_{G}+\left|V_{THP}\right|, \end{array}$$
where $V_{G}$, $V_{GSP}$, and $V_{THP}$ are the gate voltage, gate-source voltage, and threshold voltage of the power transistor, respectively. When there is a voltage difference between the bulk and source, $V_{TH}$ decreases with $V_{BS}$ due to the body effect and can be written as:(2)$$\begin{array}{c} V_{TH}=V_{TH0}+\gamma \left(\sqrt{|2\Phi_{F}|-V_{BS}}-\sqrt{|2\Phi_{F}|}\right), \end{array}$$
where $V_{TH0}$ is the threshold voltage when $V_{BS}$ is zero, γ denotes the bulk-effect coefficient, and $\Phi_{F}$ is the Fermi potential. Generally, γ ranges from 0.3 $V^{1/2}$ to 0.4 $V^{1/2}$. This analysis was simplified by replacing source-bulk voltage $V_{SB}$ with $-V_{BS}$. In the case of $a^{2}$>>$b^{2}$, taking into account a mathematical formula [24]:(3)$$\begin{array}{c} \sqrt{a+b}\approx \sqrt{a}+\frac{b}{2\sqrt{a}}. \end{array}$$

In Equation (Equation 2), there is ${\left(2\Phi_{F}\right)}^{2}$>>$V_{SB}^{2}$, and substitute them into *a* and *b*, and considering the relationship of [25,26]:(4)$$\begin{array}{c} 1+\frac{\gamma }{2\sqrt{2\Phi_{F}}}=\eta , \end{array}$$
where η ranges from 1.2 to 1.3. Part of Equation (Equation 2) can be approximately expressed as follows:(5)$$\begin{array}{c} \gamma \left(\sqrt{2\Phi_{F}-V_{BS}}-\sqrt{2\Phi_{F}}\right)\approx \frac{\gamma V_{SB}}{2\sqrt{2\Phi_{F}}}=\left(\eta -1\right)V_{SB}. \end{array}$$

Therefore, we can conclude that $V_{TH}$ is linearly related to $V_{SB}$. Via the body effect approximation, Equation (Equation 2) can be simplified as [27]:(6)$$\begin{array}{c} V_{TH}\approx V_{TH0}+\left(\eta -1\right)V_{B}. \end{array}$$

Previously, the bulk of the power transistor was connected to the source to prevent the conduction of the PN junction between the source and the bulk, as shown in Figure 2a. Thus $V_{TH}$ = $V_{TH0}$. The minimum $V_{IN}$ of this kind of LDO is larger than 1 V, which does not meet the criteria for low $V_{IN}$.

The bulk modulation is proposed to reduce $V_{TH}$ and then $V_{IN}$, as illustrated in Figure 2b. This system imposes an additional amplifier to generate the bulk voltage $V_{BODY}$, which increases power consumption.

Recently, bulk modulation without an extra amplifier was proposed in [28], which can better realize the balance between power consumption and low $V_{IN}$. Figure 2c shows its structural block diagram. However, the modulated $V_{G}$ is close to $V_{IN}$, resulting in low gain and poor performance.

To solve the problems mentioned above, this paper proposes a voltage reuse scheme to generate $V_{BODY}$. In a typical three-stage structure of OCL-LDO, the buffer is used to increase the drive ability and enhance the load capacity of the power transistor due to the small output resistance. The buffer output is usually only connected to the gate of the power transistor $V_{G}$. In this paper, the bulk modulation is simplicity obtained by connecting the buffer output to not only the gate but also the bulk of the power transistor, as shown in Figure 3. $V_{IN}$ in this paper can be expressed as:(7)$$\begin{array}{c} V_{\mathrm{IN}}=V_{G}+\left|V_{\mathrm{TH}}\right|=V_{TH0}+V_{G}+\left(\eta -1\right)V_{G}=V_{TH0}+\eta V_{G.} \end{array}$$

From Equation (Equation 7) $V_{IN}$ is determined only by $V_{TH0}$ and $V_{G}$. The large value of $V_{BODY}$ is replaced by $V_{G}$, resulting in a reduction in $V_{IN}$ to 600 mV. In addition, a gain path is established from the bulk to $V_{OUT}$, and the body transconductance $g_{mb}$ can enhance the regulation gain. Consequently, $V_{IN}$ is more effectively reduced without introducing additional power consumption, ensuring that the transistor performs normally in low $V_{IN}$.

It is important to note that a forward-biased PN junction between bulk and source experiences very low voltage. An exceedingly high $V_{SB}$ may lead to the forward breakdown of the MOS transistor, resulting in device damage. Lowering the bulk voltage appropriately, while ensuring that the PN junction does not undergo breakdown, can reduce $V_{TH}$. To validate the feasibility of the proposed approach, simulations of the circuit were conducted. Figure 4a shows that the value of $V_{TH}$ as $V_{IN}$ varies under different $V_{BODY}$ when adopting the $V_{BODY}$ reuse method. The simulation results show that $V_{TH}$ increases as $V_{IN}$ decreases, while $V_{BODY}$ remains constant. However, $V_{TH}$ can be effectively reduced by decreasing $V_{BODY}$. In the proposed design, the MOS transistor is protected from forward breakdown since the maximum value of $V_{BODY}$ is 280 mV, which is less than the forward breakdown voltage. Additionally, Figure 4b shows the drain current with and without $V_{BODY}$ reusing. There is evidence that, within a specific range, after adopting the reuse of $V_{BODY}$, $I_{Q}$ can be effectively reduced. Therefore, the proposed bulk modulation without an amplifier reduces both $V_{IN}$ and $I_{Q}$.

Despite this, bulk modulation also has some disadvantages, such as circuit instability due to the excessive modulation or reverse breakdown of PN junctions caused by a $V_{BODY}$ that is too low. To solve the above problems, APT technology is proposed in the next section.

### 2.2. Adaptive Power Transistors

The drain current $I_{D}$ of the power transistor working in the saturation region can be given by:(8)$$\begin{array}{c} I_{D}=\frac{1}{2}\mu_{P}C_{OX}\frac{W}{L}{\left(\left|V_{GS}\right|-\left|V_{TH}\right|\right)}^{2}, \end{array}$$
where $\mu_{P}$ is the mobility of the PMOS channel, $C_{OX}$ is the capacitance of the gate oxide layer, and W/L is the transistor ratio.

As the load current $I_{LOAD}$ gradually increases and $V_{GS}$ increases, so $V_{G}$ decreases accordingly. When employing $V_{BODY}$ reuse, there is $V_{G}$ = $V_{BODY}$. Reducing $V_{BODY}$ too much will result in increased $V_{BS}$, which may lead to the forward breakdown of the PN junction.

As a preventative measure, this paper proposes the APT technology. This is based on the principle that, when the load changes, a feedback signal is used to adjust the use of the auxiliary and main power transistors ($MP_{1}$ and $MP_{2}$). Meanwhile, bulk modulation performs on $MP_{1}$ and $MP_{2}$. Specifically, the modulation effect will be enhanced as $I_{LOAD}$ increases. The $V_{BODY}$ reuse scheme being implemented for $MP_{2}$ will turn off under light loads, making $V_{BODY}$ close to $V_{IN}$. The schematic diagram and working process of APT are shown in Figure 5.

The system works in a twp-stage amplifier topology when the load is light. Due to the second gain stage performing in the triode region, $M_{14}$ pulls the output voltage closer to the supply. Therefore, the $V_{GS}$ of $MP_{2}$ is minimal, and $MP_{2}$ is turned off. This was obtained by designing the aspect ratio of $M_{14}$ and $M_{16}$. $M_{14}$ is set to be small enough and $M_{16}$ is set to be large enough; thus, $M_{16}$ works in the saturation region, whereas $M_{14}$ is forced into the deep linear region to sustain equilibrium in the light load. The gate of $MP_{1}$ is connected to the output of the first-stage amplifier, $V_{EA}$. $MP_{2}$ is forcibly disabled and all the load current flows through $MP_{1}$.

When the load current increases, the $I_{DS}$ of $MP_{1}$ increases, as well as its $V_{GS}$. Therefore, $V_{EA}$ decreases, as well as the $V_{G}$ of $M_{11}$, which increases the current of $M_{11}$ and provides a large current to $M_{15}$, as well as $M_{16}$. Thus, the drop-down ability of $M_{16}$ becomes strong enough to pull $M_{14}$ from the linear region into the saturation region. Hence, $MP_{2}$ is activated. The system works in a 3-stage amplifier topology.

To better distinguish the two operating states, we define a threshold current $I_{ON}$ of approximately 350 μA. When the load current is less than $I_{ON}$, the second-stage amplifier is working in the triode region and turns off $MP_{2}$. When the load current increases to $I_{ON}$, the second-stage amplifier works in the saturation region and turns on $MP_{2}$.

It is worth mentioning that the proposed LDO can adaptively switch the power transistors according to the load current, allowing the system to transform itself between 2-stage and 3-stage topologies. In addition, simulation results have shown that $V_{BODY}$ ranges between 350 mV and 550 mV with APT, and $V_{BS}$ is less than the forward breakdown voltage, which ensures the safety of the device.

### 2.3. Structure of Proposed OCL-LDO

Figure 6 shows the LDO structure with the proposed bulk modulation and APT. The gain boost stage with an adaptive bias module (ABM) is used, which obtains a large bandwidth gain depending on the load current. The unity gain buffer (BUF) produces $V_{BODY}$ to modulate the bulk voltages of $MP_{1}$ and $MP_{2}$, lowering the total current consumption via bulk modulation. The APT not only achieves satisfactory linearity but also improves the transient response. In an attempt to realize enough stability, the compensation module (CM) with an internal RC network is used.

## 3. Circuit Implementation

This section will provide an overview of the OCL-LDO, including its structural realization and design consideration. The proposed OCL-LDO works under the condition of $V_{IN}$ = 0.6 V, $V_{REF}$ = 0.5 V, with $V_{Drop}$ = 100 mV. The LDO’s $V_{REF}$ is ultra-low, which can be generated with a BJT-based Kuijk bandgap reference [29]. Figure 7 depicts the complete circuit diagram. The gain boost stage is a transconductance-boosted amplifier ($M_{1}$–$M_{10}$) and ABM provides adaptive tail current. The BUF consists of $M_{11}$–$M_{16}$ in the form of a current mirror, which is capable of increasing the slew rate and improving the transient response. The CM is composed of a resistor $r_{m}$ and a capacitor $C_{m}$.

### 3.1. Maximize the Efficiency under Low $V_{IN}$

To maximize efficiency, it is necessary to reduce the dropout voltage. However, lowering the dropout voltage degenerates the DC gain of the power stage, which can be expressed as:(9)$$\begin{array}{cc} A_{P} & =g_{m,\mathrm{linear}}R_{OUT}\\ \; & \approx \frac{\mu_{n}C_{OX}\frac{W}{L}V_{DS}}{\mu_{n}C_{OX}\frac{W}{L}\left(V_{GS}-V_{TH}\right)}\\ \; & =\frac{V_{DS}}{V_{GS}-V_{TH}}, \end{array}$$
where $g_{m,linear}$ is the transconductance in the linear region. As $V_{DS}$ in the design is merely 100 mV; it is highly possible for it to be smaller than ($V_{GS}-V_{TH}$), implying that the $A_{P}$ is less than one. It is worth noting that this issue is not present for regular 200 mV dropout regulators, as their power stage can supply roughly 20 dB gain for the control loop. To address this problem, the proposed LDO introduces a gain boost stage with ABM.

From Figure 7, the gain boost stage employs a single-stage differential amplifier along with a cross-coupled PMOS pair to increase the gain. The cross-coupled pair generates negative resistance that can counteract the equivalent resistance of diode-connected transistors $M_{5}$ and $M_{6}$, which effectively increases the resistance of internal nodes $V_{X}$ and $V_{Y}$. The amplifier gain is obtained as:(10)$$\begin{array}{c} A_{EA}=\frac{\left(1+\alpha \right)g_{m}+g_{ds}}{\left(1-\alpha \right)g_{m}+g_{ds}}g_{m1}(r_{o8}\left|\right|r_{o10}), \end{array}$$
where $g_{m}$ represents the transconductance of $M_{3}$, $M_{4}$, and $g_{ds}$ denotes the total output conductance of node $V_{X}$ or $V_{Y}$. The gain is amplified by a factor of $\left(\left(1+\alpha \right)g_{m}+g_{ds}\right)/\left(\left(1-\alpha \right)g_{m}+g_{ds}\right)$. To ensure that it works as an amplifier rather than a hysteresis comparator, the pairs $M_{3}$-$M_{4}$ and $M_{6}$-$M_{5}$ are set at a ratio of α:1, where α should be kept smaller than 1 to avoid the over-compensation of negative resistance, taking into account the random mismatch under the fabrication.

For the push-pull stage structure, the DC gain can be approximated at 1. As a result, the overall circuit gain can be expressed as follows:(11)$$\begin{array}{c} A_{DC}=A_{P}\cdot A_{EA}=\frac{V_{DS}}{V_{GS}-V_{TH}}\cdot \frac{\left(1+\alpha \right)g_{m}+g_{ds}}{\left(1-\alpha \right)g_{m}+g_{ds}}g_{m1}\left(r_{o8}∥r_{o10}\right). \end{array}$$
Thus, the DC performance of this LDO can be significantly improved by increasing $A_{EA}$ when $A_{P}$ is small.

### 3.2. Achieve Low $I_{Q}$ and Transient Response

Realizing nA-level quiescent current is necessary to ensure ultra-high-power efficiency. The proposed LDO uses adaptive bias to ensure a low quiescent current while reducing transient response time and improving transient response.

Unlike traditional adaptive bias circuits, which mirror the current at the gate of the power transistor to bias the amplifier or buffer, the proposed ABM is driven by the first-stage amplifier itself and provides a bias current proportional to the load current, which can maintain a large enough gain of the first-stage amplifier over the entire load range.

Figure 8 illustrates the specific design of ABM. The ABM’s input current is set by $M_{23}$, which is biased by $V_{EA}$ and is proportional to the load current because $V_{EA}$ is closely related to the load current. The output current of ABM is $I_{BIAS}$, which is used to bias the amplifier and equal to the sum current of $M_{1}$ and $M_{2}$. $I_{MIN}$ and $I_{2}$ are used to define the bias upper and lower limit. The workflow of ABM can be summarized into the following three phases depending on the load current.

Light load

During periods of low load current, the current in $M_{23}$ is ultra-low and not sufficient to turn on the transistor. In this case, all devices are turned off, and $I_{BIAS}$ is equal to $I_{MIN}$.
(12)$$I_{BIAS,L}=I_{MIN}.$$

Moderate load

As the load current increases, the transistors $M_{20}$, $M_{21}$ and $M_{22}$ in the ABM are turned on, and there are
(13)$$\begin{array}{c} I_{DM20}=\frac{1}{K}\cdot I_{LOAD},\\ I_{DM21}=\frac{N}{K}\cdot I_{LOAD}. \end{array}$$

Considering that the current of $M_{20}$ is less than $I_{2}$, $I_{BIAS}$ is equal to the current sum of $M_{21}$ and $I_{MIN}$, and can be expressed as:(14)$$I_{BIAS,M}=I_{MIN}+I_{DM21}=I_{MIN}+\frac{N}{K}\cdot I_{LOAD},$$
where *K* and *N* are the current mirror ratios, as shown in Figure 8.

Heavy load

Once the current of $M_{20}$ increases to larger than $I_{2}$, the current mirror, made up of $M_{18}$ and $M_{19}$, is activated. This threshold current $I_{TH2}$ can be expressed in the following manner:(15)$$I_{LOAD}=I_{TH2}=K\cdot I_{2}.$$

$I_{BIAS}$ stops increasing and reaches the upper limit, which can be expressed as:(16)$$I_{BIAS,H}=I_{MIN}+\frac{N}{K}\cdot I_{LOAD}=I_{MIN}+N\cdot I_{2},$$

The bounds of $I_{BIAS}$ ensures stable operation. In the actual design, $I_{MIN}$ and $I_{2}$ are set at 7 nA, 18 nA, respectively. *N* and *K* are set at 6 and 5150, respectively. It is worth mentioning that ABM also contributes to improvements in gain. The equivalent transconductance $g_{mEA}$ of the gain boost stage can be determined as follows:(17)$$g_{mEA}=\frac{g_{m1}}{\frac{g_{m3}}{g_{m5}}-1},$$

As $I_{LOAD}$ increases, $g_{m1}$ increases significantly; thus, $g_{mEA}$ enlarges accordingly. Therefore, the bias current for the amplifier is proportional to the load current.

## 4. Stability Analysis

To analyze the stability of the proposed LDO, a small-signal transfer function needs to be derived. It can be seen from Figure 9 that the small-signal model switches between the two-stage amplifier and the three-stage amplifier. Due to the existence of the self-adaptive power transistor, $MP_{1}$, and $MP_{2}$ turn on and work sequentially under the condition of different load currents. Thus, the stability analysis of the circuit should be discussed according to the situation: (i) the second-stage amplifier and (ii) the third-stage amplifier.

Due to the topology complexity, it is difficult to obtain the transmission function and analyze the stability. To simplify the derivation of the loop without reducing accuracy, the following assumptions are established.

The gains of the first-stage amplifier, the main power transistor, and the side power transistor are far larger than 1, that is, $g_{m<i>}r_{o<i>}$>> 1. In the case of the third-stage amplifier, $g_{mp2}$>>$g_{mp1}$. Since the $g_{mb}$ of $MP_{1}$ is small, it was ignored in the analysis. The output capacitor $C_{L}$ is larger than compensating capacitors $C_{M}$, $C_{EA}$.

### 4.1. Two-Stage Structure

In the case of a small load current, the LDO is equivalent to a secondary structure. There is a pole in $V_{OUT}$ due to the output resistance and load capacitance. Since output resistance is closely related to the load size, a light load will have a relatively large output resistance, which results in the pole associated with the output terminal moving to the low frequency and becoming the dominant pole, $P_{1}$. The secondary pole $P_{2}$ is located at the output of the first stage because of the output resistance of the amplifier and the parasitic capacitance of the $MP_{1}$ gate. The poles’ frequencies can be obtained by calculation as follows:(18)$$\begin{array}{cc} \; & P_{1}=-\frac{1}{r_{oEA}C_{m}g_{mp1}r_{o}}, \end{array}$$(19)$$\begin{array}{cc} \; & P_{2}=-\frac{g_{mp1}}{C_{EA}}. \end{array}$$

Maintaining stability by only allowing one pole to exist within the bandwidth is necessary. When the OCL-LDO is operated under the minimum load current condition, the pole frequency located in the output is close to zero. Therefore, LDO has a very poor phase margin (PM) and must adopt a compensation structure to produce a zero point. The compensation is performed by connecting the compensation Miller capacitor $C_{M}$ between the output of the first-stage amplifier and $V_{OUT}$. However, the traditional Miller compensation produces a zero point in the right half plane (RHP), which is converted to the left half plane (LHP) by adding the zeroing resistor $r_{m}$. Therefore, the zero point $Z_{1}$ can be expressed as:(20)$$\begin{array}{cc} \; & Z_{1}=-\frac{g_{mp1}}{C_{m}\left(g_{mp1}r_{m}-1\right)}.\; \end{array}$$

Therefore, the transfer function of the two-stage structure can be expressed as:(21)$$\begin{array}{c} LG\left(s\right)=A_{0}\frac{\left(1+\frac{s}{z_{1}}\right)}{\left(1+\frac{s}{p_{1}}\right)\left(1+\frac{s}{p_{2}}\right)}, \end{array}$$
where $A_{0}$ is the DC gain under the condition of light load and can be calculated as:(22)$$\begin{array}{c} A_{0}=g_{mEA}r_{oEA}g_{mp1}r_{o}. \end{array}$$

As shown in Figure 10, the LDO’s open-loop frequency response and pole-zero distribution are plotted under the conditions of $V_{IN}$ = 0.6 V with $I_{LOAD}$ = 10 μA. Based on the simulation results, the PM is better than 45° for the worst-case scenario with the smallest load current, and the overall circuit is stable.

### 4.2. Three-Stage Structure

During heavy load conditions, the output resistance of the circuit decreases, and the pole at the output becomes the secondary pole $P_{2}$ instead of the dominant pole $P_{1}$. Since the $MP_{2}$ main power transistor has a large size, its parasitic capacitance is relatively large. Therefore, the pole existing on the gate of the main power transistor becomes the main pole $P_{1}$. The poles under heavy load can be approximated as follows:(23)$$\begin{array}{cc} \; & P_{1}=-\frac{g_{m13}}{C_{m}g_{m12}g_{m14}r_{oEA}r_{o14}}, \end{array}$$(24)$$\begin{array}{cc} \; & P_{2}=-\frac{1}{C_{L}\left(r_{mp2}∥r_{o}\right)}, \end{array}$$(25)$$\begin{array}{cc} \; & P_{3}=-\frac{g_{m13}}{2C_{gs14}}, \end{array}$$(26)$$\begin{array}{cc} \; & P_{4}=-\frac{1}{C_{g}r_{o14}}. \end{array}$$

Because the output capacitance and equivalent resistance of other nodes in the small-signal diagram are small, other poles are outside the bandwidth, which will be canceled out by zeros and thus have no effect on the stability of the overall circuit.

Due to the relationship between the equivalent capacitance of the miller compensation and the gain of the amplifier, the zero frequency changes in response and moves to a higher frequency. This zero is given by:(27)$$\begin{array}{cc} \; & Z_{1}=-\frac{g_{mp1}}{C_{m}}, \end{array}$$(28)$$\begin{array}{cc} \; & Z_{2}=-\frac{g_{m12}g_{m14}\left(g_{mb}+g_{mp2}\right)}{C_{m}g_{mp1}}. \end{array}$$

As a consequence, the three-stage structure has the following transfer function:(29)$$\begin{array}{c} LG\left(s\right)=A_{0}\frac{\left(1+\frac{s}{z_{1}}\right)\left(1+\frac{s}{z_{2}}\right)}{\left(1+\frac{s}{p_{1}}\right)\left(1+\frac{s}{p_{2}}\right)\left(1+\frac{s}{p_{3}}\right)\left(1+\frac{s}{p_{4}}\right)}, \end{array}$$
where $A_{0}$ is the DC gain under the condition of heavy load, which can be calculated as:(30)$$A_{0}=\frac{g_{mEA}g_{m12}g_{m14}\left(g_{mp2}+g_{mb}\right)r_{oEA}r_{o14}r_{o}}{g_{m13}}.$$

Figure 11 depicts the specific poles and zeros with a bandwidth under the heavy load. The figure shows that the PM is 59${}^{\circ }$, which is greater than 45${}^{\circ }$, indicating that the circuit is stable overall.

## 5. Simulation Results and Discussion

The LDO proposed in this paper was implemented and verified in a 180 nm CMOS process, which supports 0.6–1.2 V supply voltage and 100 mV dropout voltage with a 10 mA maximum load current. The system can remain stable when $C_{L}$ ranges between 0 and 1000 pF with a 10 pF on-chip compensation capacitor. When $I_{LOAD}$ = 10 μA, $V_{IN}$ varies from 0.6 V to 1.2 V, and $I_{Q}$ changes from 226 nA to 219 nA. The devices’ size and the most important parameter values for the proposed OCL-LDO are summarized in Table 1.

Under the condition of $V_{IN}$ = 600 mV, $I_{LOAD}$ increases from 10 μA to 10 mA, $I_{Q}$ changes from 220 nA to 1.2 μA. The changing trend of $I_{Q}$ in the two cases is depicted in Figure 12.

Only 1.2 μA of quiescent current is consumed when the load current is 10 mA; thus, it achieves a maximum current efficiency of 99.96% when the load current is 10 mA. The current efficiency of the LDO under full load conditions is presented in Figure 13.

The current of $MP_{1}$ and $MP_{2}$ are also shown in Figure 14. From the result, when the system works as a two-stage, system the current of $MP_{2}$ is almost equal to zero and $MP_{1}$ increases with $I_{LOAD}$. Once $I_{LOAD}$ > $I_{ON}$, $MP_{2}$ is activated, its current increases with $I_{LOAD}$ while the current of $MP_{1}$ remains unchanged.

Additionally, we simulated the relationship between PM and load current under a variety of loop conditions. As shown in Table 2, PM decreases with an increase in $I_{LOAD}$ when LDO works in a two-stage manner. However, PM in the three-stage is the opposite.

The measured DC load regulation (LDR) is shown in Figure 15a; when the load current varies from 10 μA to 10 mA, the output voltage changes by 0.05857 mV. Therefore, the LDR of the LDO is 0.00585759 mV/mA. The simulation results depicted in Figure 15b indicate that the output voltage only changes to 0.292787 mV when the supply voltage changes linearly from 0.6 V to 1.2 V. According to the calculation formula, the DC line regulation (LNR) of LDO is 0.487978 mV/V.

Figure 16a,b show the measured transient response when the OCL-LDO supply voltage is set to 0.6 V and the load capacitor is set to 100 pF, respectively. For a 1 ns edge, the maximum transient response of △$V_{OUT}$ is 230 mV, and the settling time is 5 μs. When the edge is 1 μs, the maximum △$V_{OUT}$ is 140 mV, and the settling time is 3.5 μs, which represents a significant improvement. There is a minimum overshoot voltage of 95 mV and a undershoot voltage of 140 mV. However, the transient response is not entirely satisfactory. These results indicate that the circuit’s performance is satisfactory and there is room for further improvement. Typically, an undershoot improvement circuit would be included to enhance the transient response. Nevertheless, due to the constraints of quiescent current, no improvement circuit has been incorporated.

The trend of the PSR with frequency can be plotted by analyzing the PSR at multiple frequencies over the course of one measurement. The simulated PSR performance at 10 μA load current, 0-pF $C_{L}$, and 100 mV dropout is shown in Figure 17. Hence, it can be seen that the minimum PSR is −51.40 dB, −49.17 dB at 1 Hz, and −43.97 dB at 1 k Hz.

Simulations at different temperatures and process corners are conducted to demonstrate the robustness of the proposed design. The performance results, including minimum values of PM and GM, quiescent current, LDR, LNR, transient response, current efficiency, and PSR, are presented in Table 3. The minimum GM under different corners is greater than 50 dB, ensuring high regulation performance under different conditions. $I_{Q}$ is generally within 500 nA, with the lowest value being 139 nA at the SS corner of −20 °C; the current efficiency is at least 99.95%, and the maximum is 99.99%. There is an excellent transient response at the FF corner of 27 °C, a 2.7 μs recovery time, and a maximum undershoot voltage of 320 mV. The worst PSR within 50 kHz is −51.4 dB at the TT corner of 27 °C. This shows that the proposed design can work steadily and perform as expected, even under extreme conditions. These results predict that the proposed LDO will work reliably after fabrication, even in the worst-case temperature and process corner.

A merit map (FoM) is introduced to reflect the overall performance of the LDO, which has the following formula:(31)$$FoM=K\left(\frac{C_{L}I_{Q}\Delta V_{OUT}}{\Delta I_{O,MAX}{}^{2}}\right),$$
where $I_{Q}$ is the minimum quiescent current, and $\Delta V_{OUT}$ is the maximum change in output during transient. Furthermore, *K* is the edge time ratio and defined by:(32)$$K=\frac{\Delta t\;\mathrm{used}\;\mathrm{in}\;\mathrm{the}\;\mathrm{measurement}\;}{\;\mathrm{the}\;\mathrm{smallest}\;\Delta t\;\mathrm{among}\;\mathrm{designs}\;\mathrm{for}\;\mathrm{comparison}\;}.$$

In this design, K=1, bringing the above values into the calculation, the FoM value is 0.0506 ps. The comparison of the proposed LDO’s performance with several state-of-the-art fully integrated LDOs is presented in Table 4 below. Based on the results of the analysis, the proposed LDO achieves the lowest dropout voltage, good load regulation, and lowest FoM.

## 6. Conclusions

In this paper, an nA-level and 100 mV dropout LDO, combined with bulk modulation and adaptive power transistors is proposed and simulated in the 0.18 μm CMOS process. The operation principle, circuit implementation, stability analysis, and simulation results have been presented in detail. Particular attention has been paid to obtaining a low dropout and low supply voltage by the bulk modulation technology, without an auxiliary amplifier. Adaptive power transistors are proposed to switch under different load conditions to ensure circuit stability while reducing circuit power consumption. In addition, the adaptive bias mirroring the current from the amplifier itself with bounds is utilized to improve the transient response. Due to the superiority of ultra-low-power consumption, low supply voltage, and a fast response, the proposed OCL-LDO is suitable for use as a point-of-load regulator in energy harvesting.

## Figures and Tables

**Figure 1 micromachines-14-00998-f001:**
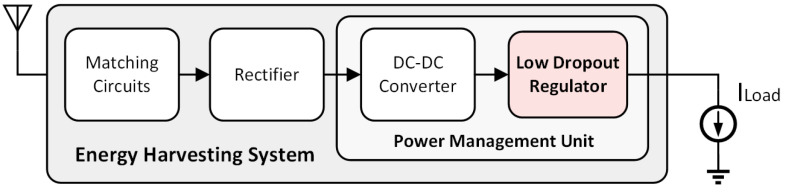
A typical EH system.

**Figure 2 micromachines-14-00998-f002:**
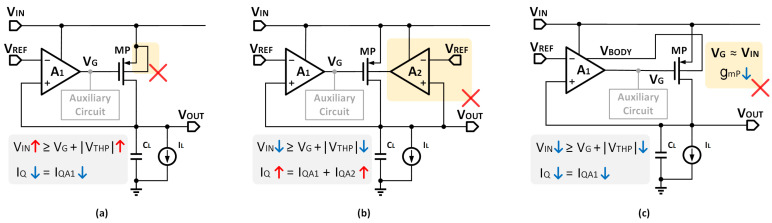
Structures of (**a**) conventional analog LDO, (**b**) bulk modulation with an extra amplifier, (**c**) bulk modulation without an extra amplifier.

**Figure 3 micromachines-14-00998-f003:**
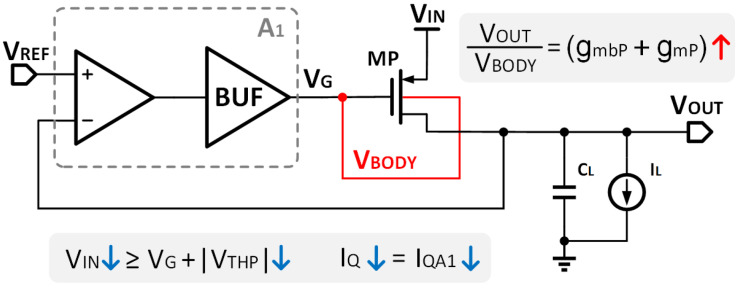
The structural diagram of the $V_{BODY}$ reuse.

**Figure 4 micromachines-14-00998-f004:**
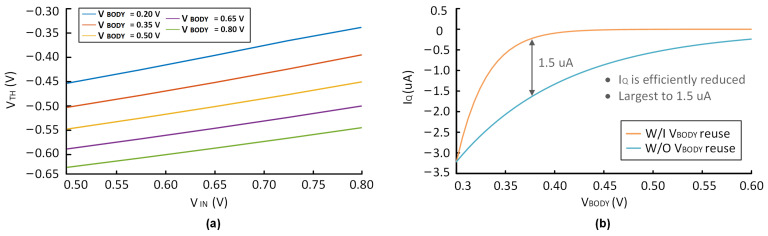
Simulation results of (**a**) the change of $V_{TH}$ with $V_{IN}$, in the case of different $V_{BODY}$, (**b**) the $I_{Q}$ with $V_{BODY}$ reuse and without $V_{BODY}$ reuse.

**Figure 5 micromachines-14-00998-f005:**
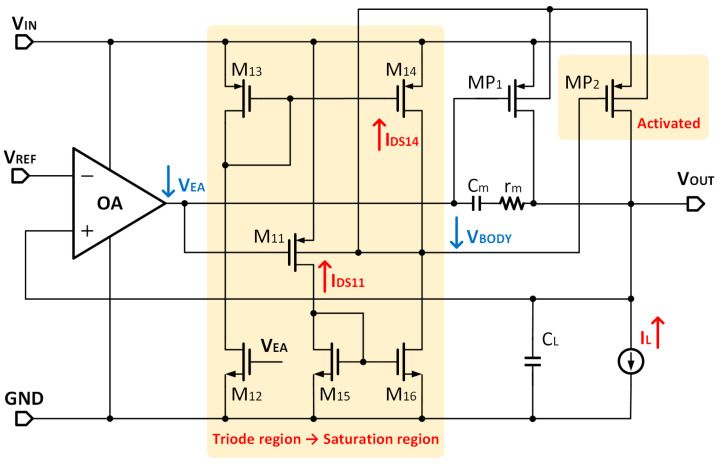
The schematic diagram and working process of APT.

**Figure 6 micromachines-14-00998-f006:**
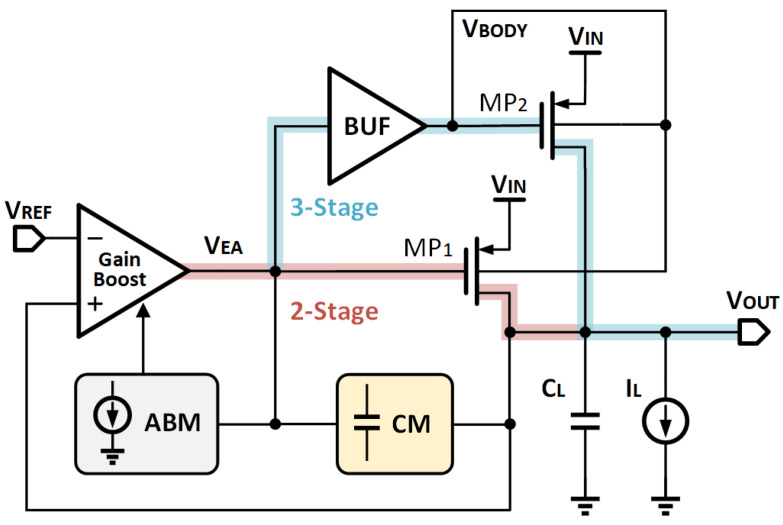
The black box schematic of the proposed LDO.

**Figure 7 micromachines-14-00998-f007:**
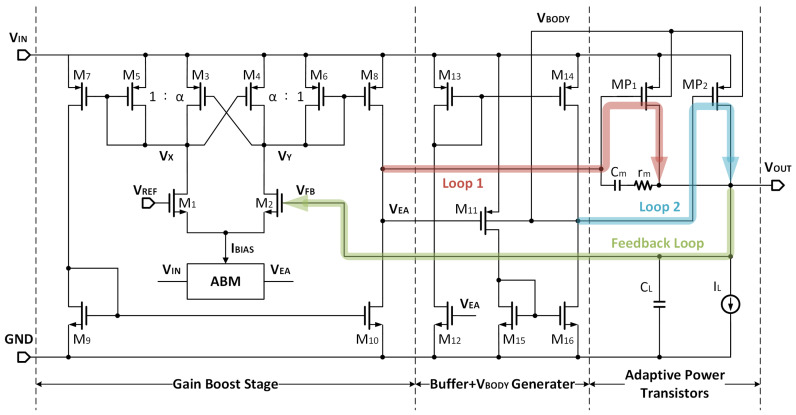
The schematic of the proposed LDO.

**Figure 8 micromachines-14-00998-f008:**
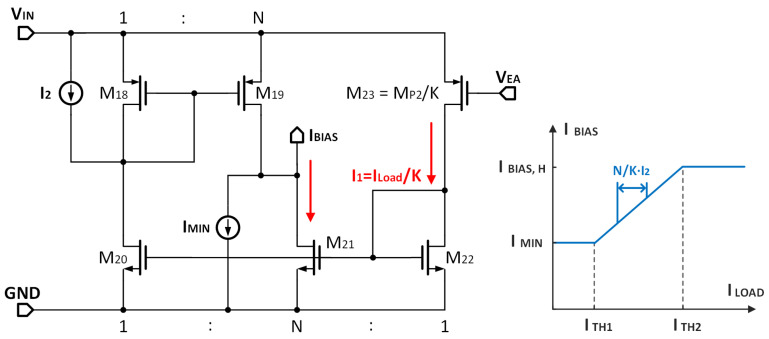
The schematic of ABM.

**Figure 9 micromachines-14-00998-f009:**
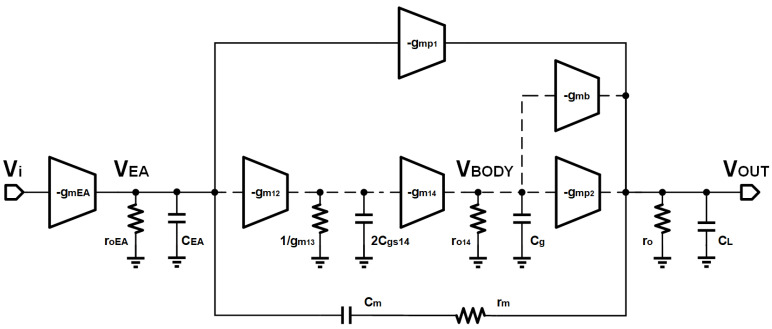
The small-signal model of the proposed LDO.

**Figure 10 micromachines-14-00998-f010:**
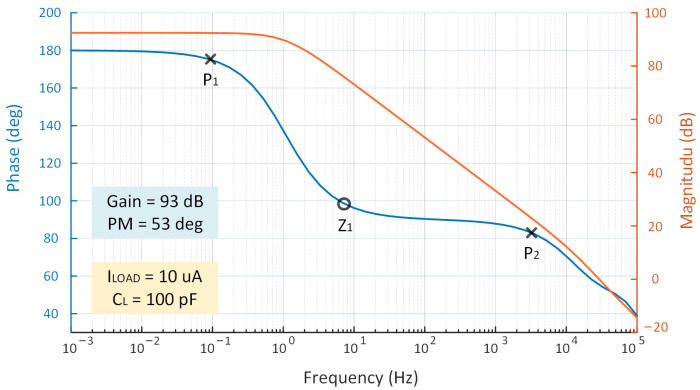
Open-loop gain simulation of the proposed LDO when $I_{LOAD}$ = 10 μA.

**Figure 11 micromachines-14-00998-f011:**
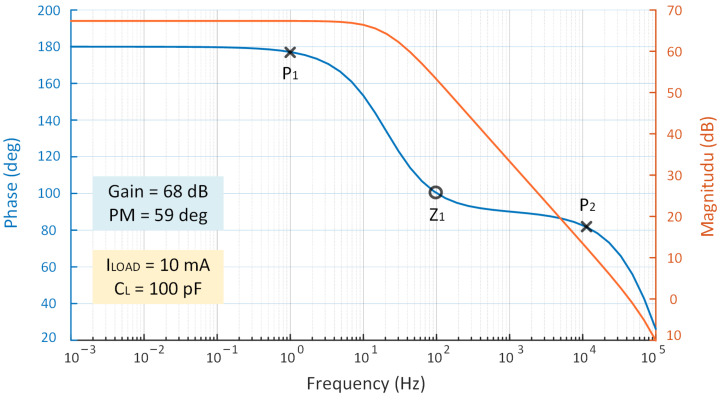
Open-loop gain simulation of the proposed LDO when $I_{LOAD}$ = 10 mA.

**Figure 12 micromachines-14-00998-f012:**
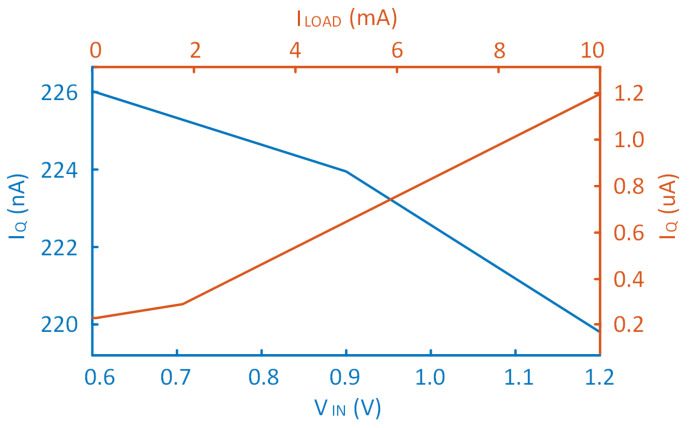
The $I_{Q}$ of the proposed LDO when $V_{IN}$ varies from 0.6 V to 1.2 V, $I_{LOAD}$ changes from 10 μA to 10 mA.

**Figure 13 micromachines-14-00998-f013:**
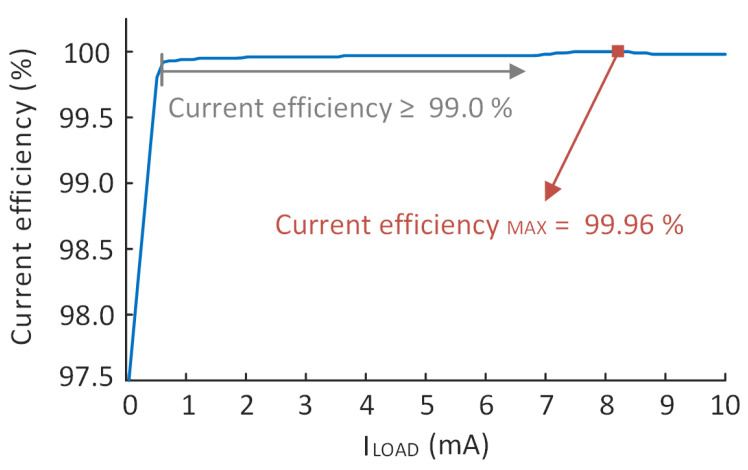
The current efficiency of the proposed LDO when $I_{LOAD}$ changes from 10 μA to 10 mA.

**Figure 14 micromachines-14-00998-f014:**
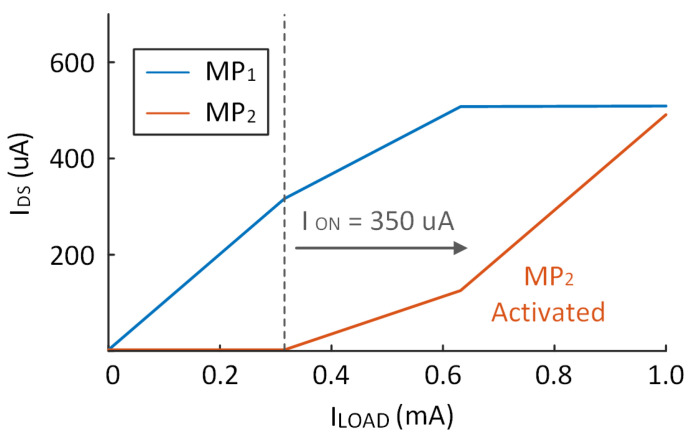
$I_{D}$ of $MP_{1}$ and $MP_{2}$ when $I_{LOAD}$ changes from 0 to 1 mA.

**Figure 15 micromachines-14-00998-f015:**
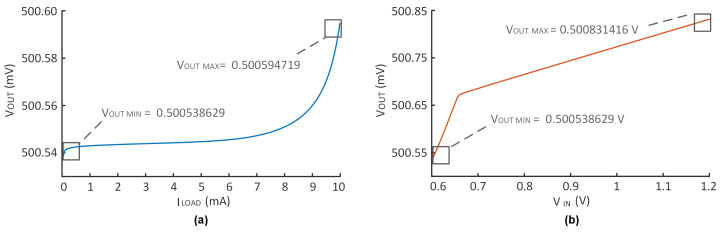
Simulation results of (**a**) DC load regulation, (**b**) DC line regulation.

**Figure 16 micromachines-14-00998-f016:**
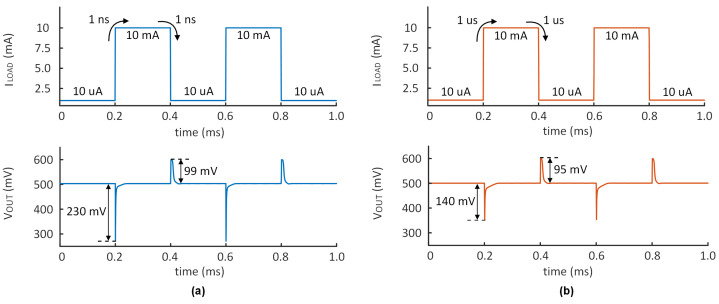
Measured dynamic response with a (**a**) 10 μA to 10 mA load step and 1 ns rising/falling edges, (**b**) 10 μA to 10 mA load step and 1 μs rising/falling edges.

**Figure 17 micromachines-14-00998-f017:**
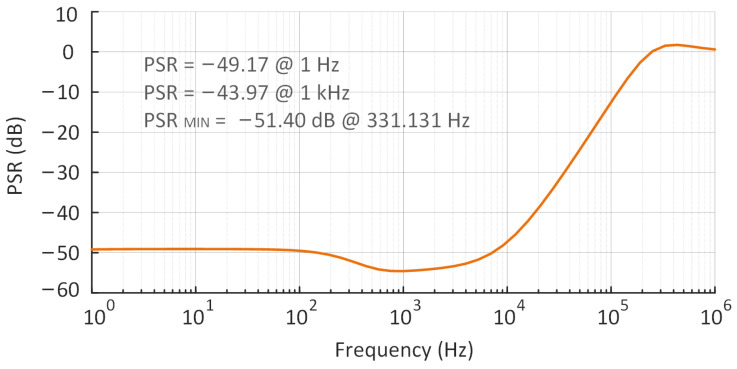
The simulated PSR performance of the proposed LDO at 10 μA load current, 0 pF $C_{L}$, and 100 mV dropout.

**Table 1 micromachines-14-00998-t001:** Devices’ size and parameters’ value of proposed OCL-LDO.

Component	Size	Component	Size	Parameters	Value
$M_{1}$, $M_{2}$	5 μ/1 μ	$M_{12}$	500 n/5 μ	α	0.75
$M_{3}$, $M_{4}$	3 μ/1 μ	$M_{13}$	500 n/500 n	N	6
$M_{5}$, $M_{6}$, $M_{7}$	4 μ/1 μ	$M_{14}$	3.5 μ/500 n	K	5150
$M_{8}$, $M_{9}$	2 μ/1 μ	$M_{16}$	250 n/1 μ	$I_{MIN}$	7 nA
$M_{10}$, $M_{15}$	1 μ/1 μ	$M_{P1}$	70 μ/180 n	$I_{2}$	18 nA
$M_{11}$	1 μ/2 μ	$M_{P2}$	20 m/180 n		

**Table 2 micromachines-14-00998-t002:** PM with respect to $I_{LOAD}$ in the two working conditions.

Loop	2-Stage			3-Stage		
$I_{LOAD}$ (A)	100 μ	200 μ	300 μ	1 m	5 m	9 m
PM (deg)	49.1	47.8	44.5	51.1	55.4	58.2

**Table 3 micromachines-14-00998-t003:** Performance summary under process and temperature corners.

Parameter	27 °C		−20 °C	80 °C	
Corner	TT	FF	SS	FF	SF
$Gain_{MIN}$ (dB)	68.4	65.1	78.51	71.1	53.4
$PM_{MIN}$ (deg)	53.64	54.98	64.38	53.85	63.33
$I_{Q}$ (nA)	220	370	139	870	424
LNR (mV/V)	0.48	0.78	0.60	0.83	0.73
LDR (μV/mA)	5.85	10.21	15.87	48.87	152.31
△$V_{OUT}$ (mV)	230	320	346	465	428
Settling Time (μs)	3.8	2.7	7.1	5.3	13.1
Current Efficiency (%)	99.96	99.99	99.99	99.95	99.95
Min. PSR from DC to 50 kHz (dB)	−51.4	−63.4	−63.7	−55.9	−62.2

**Table 4 micromachines-14-00998-t004:** Performance comparison with other works.

Paper	TCAS-I [30]	TPE [31]	TPE [32]	AEU [33]	MDPI [11]	MDPI [10]	This Work
Year	2018	2018	2020	2021	2022	2022	2023
Technology (nm)	65	130	65	180	40	40	180
$V_{IN}$ (V)	1	1	0.95	1.3	1.1	1.1	0.6
$V_{OUT}$ (V)	0.8	0.8	0.8	1.1	0.9	0.9	0.5
Vdrop (mV)	200	200	150	200	200	200	100
Capacitor-less	Yes	Yes	Yes	Yes	Yes	Yes	Yes
$I_{LOAD(max)}$ (mA)	25	100	100	100	100	100	10
$C_{L}$ (pF)	0–25 p	0–25 p	0–100 p	0–100 p	0–100 p	0–100 p	0–1000 p
$I_{Q}$ (μA)	24.2	112	14	50.25	24.6-65	30	0.22
Line Reg. (mV/V)	0.7	2.25	12	0.75	N.A.	0.2	0.487
Load Reg. (mV/mA)	0.28	0.173	0.09	0.48	0.017	0.25	0.00585
Edge time (ns)	100	10	220	500	100	100	1
Edge time ratio K	100	10	220	500	100	100	1
△$V_{OUT}$ (mV)	71	35	230	336	33	23.5	230
Min. PSR from DC to 50 kHz (dB)	−11	−22	−33	−22.5	−46	−70	−51.4
FoM (ps)	6.872	0.098	7.084	84.42	0.8118	0.705	0.0506

## Data Availability

Not applicable.

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
