# Peer review of "A 0.6 VIN 100 mV Dropout Capacitor-Less LDO with 220 nA IQ for Energy Harvesting System"

_micromachines, 2023, doi:10.3390/mi14050998_

Round 1
Reviewer 1 Report
This is an interesting topic. However, I found a question about design, performance, and overall presentation issues. The following comment needs to be considered carefully for your work.
1. Please explain in detail the derivation process from Eq.2 to Eq.3, and explain the definition of parameters and the value of η in the formula of Eq.3.
2. The parameters described in the text and the formula should be consistent with the expression results of the internal parameters in the figure (for example: in line 89 mentions the lower VBODY, the lower VTH is. and the parameters in Eq.4 are inconsistent with the expression results in Figure 4.)
3. According to the point of view described by the author, the lower the bulk voltage, the better, but for MOS, the lower the bulk voltage, the effect is forward breakdown. Please elaborate why.
4. Please clearly describe the connection mode and operating principle of M11, MP1, and MP2 in Figure 5, and make corrections and explanations..
5. 6. In Figure 8, the current of M19 under heavy load should be N*I2+IBias, please correct the description in Eq.11.
Reviewer 2 Report
This manuscript describes a low-drop capacitor-less LDO for energy harvesting system.
The proposed LDO has a low drop output voltage although using a supply voltage of 0.6V. However, the delta output voltage is relatively large as 230 mV. Also, it has a small maximum load current of 10 mA.
It is hoped that improvements for these shortcomings will be discussed in the revised manuscript.
There is no critical issue.
Reviewer 3 Report
The authors present an LDO for energy harvesting systems with very low quiescent current consumption and capable of working with supply voltage as low as 0.6V. The paper is well organized and well written, the proposed LDO topology is quite interesting and shows good results. I have some minor comments to help the authors improving their work:
- Please define eta in eq. 3.
- In the introduction, when the authors discuss the methods to improve the efficiency, the argumentation can be misleading. In fact, it is not evident if the specification is the dropout voltage or Vout. I understand that is important to minimize the dropout voltage, but usually Vout is the real specification that can be dictated by the supplied circuits. Please clarify better that part.
- The authors never mention how Vref is generated. I know that is above the scope of this work, but few words about a possible realization of the reference voltage could be useful, for example in the introduction. For example, you can consider these references: "Ria A. et al., A Low-Power CMOS Bandgap Voltage Reference for Supply Voltages Down to 0.5V", Electronics 2021", "Chi-Wa, U. et al., A 0.5V Supply, 36nW Bandgap Reference With 42 ppm/°C Average Temperature Coefficient Within -40°C to 120°C", IEEE TCAS1".
- In section 3.1, it could be useful to show the loop gain as a function of AP, AEA in order to better explain the issues.
- I suggest the authors to include a table with the sizing of the transistors and discuss the most important parameters (for instance, the value of alfa, or how N and K have been chosen).
- Fig.7 could be improved, adding that ABM is also connected to VEA and VIN, and which current is IBIAS.
- It would be very interesting a graph pf the margin phase with respect to the load current in the two working conditions (2-stage and 3-stage loop).
- It's not very clear the value of the settling time of Vout with respect of the load steps of Fig.16. In the text the authors say "the settling time is 1ns and 1us when the load current changes from 10 uA to 10 mA, respectively", but from Fig.16 it seems that 1ns a 1us are the rising and falling edges. Please clarify.
Please doublecheck because there are some minor spelling mistakes.
Round 2
Reviewer 1 Report
Thanks for replying. All questions are answered.